# Three weeks of a home-based "sleep low-train low" intervention improves functional threshold power in trained cyclists: A feasibility study

**Samuel Bennett**[1,2], **Eve Tiollier**[2], **Franck Brocherie**[2], **Daniel J. Owens**[1], **James P. Morton**[1], **Julien Louis**[1] *

1 Research Institute for Sport and Exercise Science (RISES), Liverpool John Moores University, Liverpool, United Kingdom, 2 Laboratory Sport, Expertise and Performance (EA 7370), French Institute of Sport, Paris, France

* J.B.Louis@ljmu.ac.uk

## Abstract

**Data Availability Statement:** All relevant data are within the paper.

### Background

"Sleep Low-Train Low" is a training-nutrition strategy intended to purposefully reduce muscle glycogen availability around specific exercise sessions, potentially amplifying the training stimulus via augmented cell signalling. The aim of this study was to assess the feasibility of a 3-week home-based "sleep low-train low" programme and its effects on cycling performance in trained athletes.

### Methods

Fifty-five trained athletes (Functional Threshold Power [FTP]: 258 ± 52W) completed a home-based cycling training program consisting of evening high-intensity training (6 × 5 min at 105% FTP), followed by low-intensity training (1 hr at 75% FTP) the next morning, three times weekly for three consecutive weeks. Participant's daily carbohydrate (CHO) intake (6 g·kg$^{-1}$·d$^{-1}$) was matched but timed differently to manipulate CHO availability around exercise: no CHO consumption post- HIT until post-LIT sessions [Sleep Low (SL), n = 28] or CHO consumption evenly distributed throughout the day [Control (CON), n = 27]. Sessions were monitored remotely via power data uploaded to an online training platform, with performance tests conducted pre-, post-intervention.

### Results

LIT exercise intensity reduced by 3% across week 1, 3 and 2% in week 2 ($P < 0.01$) with elevated RPE in SL vs. CON ($P < 0.01$). SL enhanced FTP by +5.5% vs. +1.2% in CON ($P < 0.01$). Comparable increases in 5-min peak power output (PPO) were observed between groups ($P < 0.01$) with +2.3% and +2.7% in SL and CON, respectively ($P = 0.77$). SL 1-min PPO was unchanged (+0.8%) whilst CON improved by +3.9% ($P = 0.0144$).

**Funding:** The authors received no specific funding for this work.

**Competing interests:** The authors have declared that no competing interests exist.

## Conclusion

Despite reduced relative training intensity, our data demonstrate short-term "sleep low-train low" intervention improves FTP compared with typically "normal" CHO availability during exercise. Importantly, training was completed unsupervised at home (during the COVID-19 pandemic), thus demonstrating the feasibility of completing a "sleep low-train low" protocol under non-laboratory conditions.

## Introduction

Muscle glycogen can mediate cell signalling pathways associated with endurance training adaptation [1], inducing an augmented muscle transcriptional response when exercise is completed under conditions of reduced muscle glycogen availability [2–6]. Indeed, when compared with loaded glycogen stores, training with reduced muscle glycogen has been shown to increase AMP-activated protein kinase (AMPK) activity as a result of decreased AMPK-glycogen binding [7, 8]. AMPK therefore acts as a cellular energy sensor, upregulating peroxisome proliferator-activated receptor 1 coactivator alpha (PGC-1α) activity and expression [1, 9, 10], a transcriptional co-activator often touted as the master regulator of mitochondrial biogenesis [11–13], a key hallmark of endurance training adaptation [14–16]. Concomitant with these adaptations, the increased body fat mobilisation for energy supply during exercise with low glycogen availability, upregulates peroxisome proliferator-activated receptor (PPARδ) transcription factor [17], thus increasing the expression of proteins involved in lipid metabolism. Such metabolic adaptation may be beneficial to improve performance during prolonged submaximal steady state exercises, via sparing of glycogen stores for a later utilisation [18, 19]. Accordingly, over the last decade, various exercise-dietary carbohydrate (CHO) periodisation strategies (*i.e.*, twice a day training, fasted training, withholding CHO intake between exercise sessions) to train with low muscle glycogen (coined as "train low") have been tested in athletes [4, 5, 20–26]. However, despite growing evidence of the molecular adaptation triggered by such "train low" strategies, the translation to improved physical performance remains limited [20, 22, 26, 27].

In the context of nutrition and exercise prescription for athletes, an amalgamation of multiple "train low" strategies appears optimal as it can be tailored to individual requirements throughout a training cycle [28, 29]. A popular example of this approach is the so called "sleep low-train low" strategy, which includes three different training-nutrition interventions: high-intensity training (HIT) in the evening to deplete glycogen stores, followed by low CHO availability overnight (i.e., sleeping low), and low-intensity training (LIT) the next morning under conditions of low muscle glycogen/CHO availability. The "sleep low-train low" model seems particularly adapted to athletic populations because the timing of exercise and CHO restriction minimises waking hours and maximises the duration under low CHO conditions, potentially maximising the adaptive response [17, 30]. Marquet et al. [23] advanced the "sleep low-train low" model to closer reflect "real world training practices" as CHO availability is periodised to suit specific exercise session demands. In a cohort of trained triathletes, these researchers observed that a 3-week "sleep low-train low" intervention improved body composition (-1.1% fat mass), 10-km running performance (- 2.9%) and submaximal cycling efficiency (+11.7%) when compared to training in conditions of consistently high CHO availability. On the contrary, using a similar "sleep low-train low" design over 4

weeks, Riis et al. [24] reported no superior effects to endurance performance in cyclists compared to a control condition. Whilst both studies provided comprehensive pre- and post-intervention performance data, each failed to report the sessional impact (i.e., ability to complete the desired workload) of training chronically with periodised CHO availability. Given that reduced CHO availability negatively impairs exercise capacity and may impact session quality [5, 31, 32], it is important to further characterise the daily training response associated with low CHO availability.

Laboratory-based training studies are often limited by logistical constraints and impose a burden on participants who must report daily to the research facility. Additionally, some sessions are often completed away from the laboratory environment, thus reducing the opportunity for data collection by the research team. Within this context, home-based studies that allow participants to train at home and collect their own data, present an alternative to researchers. Nowadays, several commercially available online training platforms are available to athletes and coaches. These packages allow athletes to record and upload exercise data and provide general and/or specific subjective feedback for each training session, ultimately allowing the coach to monitor training programme effectiveness without meeting the subject in person. The interest for this training monitoring solution has grown significantly among athletes and coaches in 2020 (likely due to the worldwide COVID-19 pandemic) with a 400% increase in virtual sessions uploaded to the TrainingPeaks (TrainingPeaks, LLC. CO, USA) platform in May 2020 compared to 2019.

The aim of the present study was to assess the feasibility of undertaking a 3-week home-based "sleep low-train low" programme and measure its effects on cycling performance in trained athletes. We hypothesise that the battery of home-based performance tests will be reliable, and performance will be improved in the Sleep Low (SL) group compared to control in line with current sleep low literature.

## Methods

### Participants

Seventy-one trained cyclists and triathletes initially volunteered to participate in the present study, following an online social media recruitment drive. Sample size calculation indicated a minimum of 16 subjects to detect a 5% difference between groups at 95% statistical power. Ten participants withdrew citing changes in work, lockdown conditions and/or illness. A further six participants were excluded from the study for failing to adhere to the training programme as prescribed (100% session completion was required). Consequently, a total of fifty-five (47 males and 8 females) trained cyclists and triathletes were considered for the analysis. Subjects were classified as trained based on training status and modelled $VO_{2max}$ PPO based upon a linear regression model of FTP and $P_{max}$ [33] against previous characterisation frameworks [34–36]. Participants were matched based upon sex, age, and functional threshold power (FTP) before being randomly assigned to either sleep low (SL, n = 28) or control (CON, n = 27) groups using a simple random allocation approach [37]. All participants were in good health, not habitually consuming metabolism altering supplements or medication, had a minimum of 2 years' experience in either cycling or triathlon and a minimum of 10 h of weekly training volume in the year preceding the study. Participants' characteristics are summarized in Table 1. The experimental protocol was approved by Liverpool John Moores University research ethics committee (Ethics number 20/SPS/019) and performed in line with the declaration of Helsinki. After comprehensive written and verbal (online) explanations of the study, participants gave their informed consent for participation.

**Table 1. Participants' characteristics in the sleep low and control groups, 12:3 male to female ratio in each group.** (mean ± SD).

| | Sleep low (SL) | Control (CON) |
|---|---|---|
| | (n = 28) | (n = 27) |
| Age (years) | 32 ± 8 | 32 ± 8 |
| Height (cm) | 177 ± 6 | 177 ± 9 |
| Weight (kg) | 75 ± 18 | 77 ± 19 |
| FTP (W) | 255 ± 53 | 258 ± 52 |
| Hours of training (h·wk$^{-1}$) | 12 ± 3 | 13 ± 4 |

FTP, Functional Threshold Power.

## Study overview

In a progression towards greater real-world application, all exercise and dietary interventions were prescribed and monitored remotely using a commercially available online training platform, and performance measures were typical of field-based tests employed by coaches in practice. During a period of government enforced confinement in 2020, a remote prescription training study was implemented utilising a matched pairs design. Following an initial familiarization, both groups completed 2 pre- and post-performance tests, with pre- tests completed twice in the same week to allow for an assessment of test battery reliability. Pre and Post tests were separated by an identical 3-week exercise intervention, which consisted of an evening HIT session, followed by a LIT session the next morning, three times per week. Both groups had the same relative daily CHO intake (~ 6 g·kg BM$^{-1}$·d$^{-1}$) but timed differently between groups to achieve low CHO availability around specific training sessions in SL, and normal CHO availability in CON. This cycle was repeated 3 times during 4 consecutive days, before participants could resume their usual dietary intake as recorded prior to the study (described later in "Nutritional Protocol").

## Performance tests

In an attempt to mirror the reality of online coaching and training practices, common field/home-based cycling tests were used to assess performance (*i.e.*, 20-min, 5-min and 1-min peak power output [PPO]) [38], via online software (TrainingPeaks, CO, USA). Participants were asked to complete the exercise tests at the same time each day, in a well hydrated state, having consumed the same foods and abstained from alcohol, caffeine and vigorous exercise for the 24 hours prior. A standardized meal (CHO: 2.0g·kg·BM, PRO: 0.3g·kg·BM, fat: 0.3g·kg·BM) 2 hours prior to the test and water consumption was allowed *ad libidum* during. Upon completion, participants uploaded power data to the online software to be analysed by the research team.

## Assessment of functional threshold power (FTP)

On the morning of the scheduled FTP test, participants were asked to weigh themselves nude, using their own scales and input the data manually to the online software. Participants were asked to complete the exercise tests at the same time each day and consume the same meal at least 2 h prior to commencing the test. Participants then completed a standardised 15-min warm-up consisting of 5-min cycling at 100 W, followed by 5-min incremental cycling, increasing by 25 W each minute until 225 W is reached. Two minutes of self-selected recovery was completed prior to completing three 10-s sprints separated by 50 s of recovery at a

self-selected intensity. Following the warm-up, participants were asked to commence the 20-min maximal effort, during which they were advised to maintain the highest possible power for the duration of the test.

FTP (W) was calculated as 95% of the mean power achieved for 20 min [38]. FTP is the highest theoretical steady state power output a cyclist can maintain for approximately 60 min [39] and has been shown to be a positive predictor of cycling performance [40], and a better predictor of mass start cycling performance than $VO_{2max}$ [41]. In light of this study being completed remotely, all participants completed the test at home, FTP was used to standardise relative exercise intensity for training sessions in place of $VO_{2max}$ [33].

## Assessment of 1-min and 5-min peak power output

Participants completed the same standardised warm-up as prior to the FTP test, following which they were asked to complete the 1-min maximal cycling effort, with the aim of achieving the highest mean power possible. Participants completed 10-min active recovery at a self-selected intensity, then completed the 5-min maximal effort. As with the FTP test, a familiarisation session was prescribed in the week prior to the test to all participants for an attempt at the maximal efforts prior to the pre-test. The tests were repeated between 4 and 7 days after the final training session (Post-tests) to allow adequate recovery from the training load.

## Training protocol

Following cycling performance tests, all participants completed a 3-week online-supervised training programme. During the 3 consecutive weeks, participants followed the same exercise programme (Table 2), however each group followed a different nutritional intervention (subsequently described in detail). The training programme was comprised of 6 sessions over 4 days, including HIT sessions in the evenings, and LIT the following mornings. The training intensity was standardized across all participants and prescribed relative to FTP. LIT sessions consisted of 60 min of cycling at 75% FTP (mean ± SD for all participants: 192 ± 41 W), whilst HIT sessions consisted of 6 × 5 min cycling at 105% FTP (269 ± 58 W) interspersed with 5-min recovery at 55% FTP (141 ± 30 W). Eight 5-min bouts at high-intensity with 60 s recovery [42] has shown to be effective at significantly reducing muscle glycogen content (~50%) and was used in a "sleep low-train low" intervention previously [23]. All sessions were structured on the online training platform and could be exported and completed on third-party applications in ergometer mode on the participants' own home trainers. Low intensity exercise was permitted on the other 3 days of each week but was limited to 1–1.5 h per day for a total weekly training volume of 10–15 h and was also monitored. Participants performed all the exercise sessions at home, using their own equipment and recording all their own data. Power

**Table 2. Overview of prescribed exercise (in bold) and CHO intake (g·kg⁻¹·BM⁻¹) during 1-week of the intervention for sleep low (SL) and control (CON) groups.**

| TIME | DAY 1 | | DAY 2 | | DAY 3 | | DAY 4 | | DAY 5–7 | |
|---|---|---|---|---|---|---|---|---|---|---|
| GROUP | CON | SL | CON | SL | CON | SL | CON | SL | CON | SL |
| MORNING (BEFORE 10AM) | Breakfast (2 g.kg⁻¹) | | Breakfast (2 g.kg⁻¹) | **LIT** (Fasted) | Breakfast (2 g.kg⁻¹) | **LIT** (Fasted) | Breakfast (2 g.kg⁻¹) | **LIT** (Fasted) | 1 free **LIT** session per day | |
| | | | **LIT** | Breakfast (2 g.kg⁻¹) | **LIT** | Breakfast (2 g.kg⁻¹) | **LIT** | Breakfast (2 g.kg⁻¹) | | |
| MIDDAY | Lunch (1.5 g.kg⁻¹) | Lunch (2 g.kg⁻¹) | Lunch (1.5 g.kg⁻¹) | Lunch (2 g.kg⁻¹) | Lunch (1.5 g.kg⁻¹) | Lunch (2 g.kg⁻¹) | Usual Diet | | Usual diet | |
| AFTERNOON (BEFORE 5PM) | Snack (0.5 g.kg⁻¹) | Snack (2 g.kg⁻¹) | Snack (0.5 g.kg⁻¹) | Snack (2 g.kg⁻¹) | Snack (0.5 g.kg⁻¹) | Snack (2 g.kg⁻¹) | | | | |
| EVENING (AFTER 5PM AND BEFORE 9PM) | **HIT** | | | | | | | | | |
| | Dinner (2 g.kg⁻¹) | Dinner (0 g.kg⁻¹) | Dinner (2 g.kg⁻¹) | Dinner (0 g.kg⁻¹) | Dinner (2 g.kg⁻¹) | Dinner (0 g.kg⁻¹) | | | | |

HIT, high intensity training session; LIT, low intensity training session.

**Table 3. Distribution of power meter utilisation by participants and literature to support validity and reliability of each device.**

| Power meter brand | Count | Strain Gauge location | Claimed Accuracy* | Literature for Validity and Reliability |
|---|---|---|---|---|
| Wahoo | 16 | Home Trainer | 1.50% | [44] |
| TACX | 13 | Home Trainer | 1% | |
| Powertap | 6 | Pedal | 1.50% | [45–50] |
| Stages | 6 | Crank | 1.50% | [45, 48, 50–52] |
| Assioma | 4 | Pedal | 1% | [53] |
| Wattbike | 4 | Home Trainer | 2% | [54] |
| Quarq | 4 | Crank | 1.50% | [45, 48] |
| Garmin Vector | 2 | Pedal | 1% | [45, 50, 55] |

* Manufacturer's claimed power meter accuracy.

output was available for all participants via a power meter or smart trainer and heart rate (HR) was recorded by those who had a thoracic HR belt (n = 48; SL = 25, CON = 23) which were either Polar (Polar Electro Oy, Kempele, Finland) (n = 21) or Garmin (Garmin International, Kansas City, MO) (n = 27) heart rate monitors. Heart rate data is reported as a percentage of predicted heart rate max, calculated as described by Tanaka et al., [43]. Participants reported their power meter brand and model to the research team and were instructed to complete a factory calibration for power meters and spin-down calibration for home trainers at the start of each week of the study. Characteristics of power meters (including brand, location of strain gauge, and data accuracy) utilised by participants can be found in the Table 3.

Immediately post exercise, participants were asked to upload data files to the online platform and rate their perceived exertion for the entire session via the online software (within 15 minutes of exercise completion. For ease of understanding for participants, a BORG CR-10 scale [56] was used in line with the available RPE scale housed within the online platform. Participants were also provided with an electronic version of the Borg CR-10 scale including written cues for the level of exercise [57]. All participants' training sessions were prescribed at the start of the study and overseen throughout by a central coach account where data could be exported for analysis.

## Nutritional protocol

During the familiarisation week, participants were asked to record their dietary intake for the duration of the week. Participants recorded all food and fluid intake using a nutrition analysis application (MyFitnessPal inc, CA, USA). Participants were asked to weigh their food prior to any preparation or cooking and asked to input the food and quantities into the nutrition analysis application each day. Nutritional data synchronised daily macronutrient intake to the online training platform. This served as a familiarisation of the nutritional monitoring that would be used during the study, and mean macronutrient intake during the final 4 days was used to calculate baseline dietary intake. Participants were asked to follow prescribed meal plans that consisted of the same total macronutrient intake, the only difference being the timing of intake to ensure contrasting periodization of CHO over the course of the day depending on their allocated group. Total daily CHO was the same for each group (6 g·kg $BM^{-1}$·$d^{-1}$), but the intake was distributed differently throughout the day to achieve either low (SL) or normal (CON) CHO availability around specific training sessions (Table 3). Across the 4 days per week that the "sleep low-train low" model was implemented, CHO consumption was prohibited in the period between the completion of HIT and LIT sessions in SL. No CHO was consumed during any training session, and to maintain satiety participants were allowed to

consume high protein food sources such as lean meat or whey protein powder in line with daily macronutrient intake. A high CHO diet was consumed upon completion of LIT sessions to replenish muscle glycogen prior to the following HIT session. For CON, CHO availability was maintained throughout the day, including post-HIT session and prior to completion of the LIT session. All participants were instructed to consume 2 g·kg$^{-1}$ BM.d$^{-1}$ of protein to maintain muscle protein synthesis. To ensure that the meal plan was followed, nutritional intake was monitored across the 4 consecutive days of the exercise-nutrition intervention.

### Statistical analysis

Data analysis was completed using Graphpad Prism v.9 (Graphpad Software, CA, USA) unless otherwise stated. To assess the difference between repeated trials, within-participants t-tests were completed for each measure (20-min, 5-min, 1-min PPO tests [mean power and heart rate] and FTP) with comparisons of reliability completed by calculating mean difference, effect size (Cohen's d), coefficient of variation (CV), typical error of the mean (TEM) and intraclass correlation (ICC) using a spreadsheet provided by Hopkins [58]. For each pair of data, including test and retest, a simple linear regression was fitted, and the coefficient of determination ($r^2$) was calculated. Linear regressions were performed between each 20-, 5-, 1-min PPO tests and calculated FTP.

 To assess the effect of intervention and any difference between experimental groups (SL *vs.* CON), an independent t-test was performed at baseline. Normal distribution was systematically checked using Shapiro-Wilk's test. A repeated measures two-way (group × time) ANOVA was used to assess the effects of the dietary strategy (SL *vs.* CON) and time (pre *vs.* post, and week 1 *vs.* week 2 *vs.* week 3) on performance outcomes (20-min PPO, 5-min PPO, 1-min PPO) and training responses (HR, RPE). A Bonferroni multiple comparisons test was performed when a significant effect was found. Degrees of freedom were adjusted using the Greenhouse-Geisser correction when violations of sphericity were present. Cohen's *d* coefficient for effect size was also calculated and referenced against benchmarks suggested by Cohen [59], where *d* is considered small, medium, and large for 0.2, 0.5 and 0.8 values, respectively. All data are presented as mean ± SD, unless otherwise stated. The level of significance was set at $P < 0.05$.

## Results

### Training response

Total weekly training volume was comparable between groups with 12.9 ± 0.7 h and 13.4 ± 1.2 h in SL and CON, respectively ($P = 0.06$). This was in line with the instruction to maintain typical weekly training hours and represented only a small increase from habitual training loads with SL completing an additional 1 hour of exercise per week ($P > 0.05$), and CON completing an additional 0.8 hours ($P > 0.05$).

 Mean absolute power output recorded across all LIT (SL: 192 ± 40W; CON: 194 ± 39W) and HIT (SL: 269 ± 56W; CON: 271 ± 54W) sessions was not different over time ($P = 0.26$) or between groups ($P = 0.42$). When normalised to FTP (Fig 1A), mean power for SL during LIT sessions was systematically lower than the 75% FTP target intensity, and significantly lower than CON in session 1 (SL: 72 ± 4% *vs.* CON: 76 ± 3%, $P \leq 0.01$, $d = 1.08$), session 2 (SL: 72 ± 5% *vs.* CON: 75 ± 2%, $P = 0.01$, $d = 0.93$), session 3 (SL: 72 ± 4% *vs.* CON: 75 ± 2%, $P = 0.01$, $d = 0.92$) and session 9 (SL: 72 ± 4% *vs.* CON: 75 ± 3%, $P = 0.01$, $d = 0.95$). On the contrary, there was no between-group difference ($P = 0.26$) in mean power output (expressed as % FTP) achieved during high intensity intervals of HIT sessions across the intervention ($P = 0.30$, $d = 0.37$) (Fig 1B).

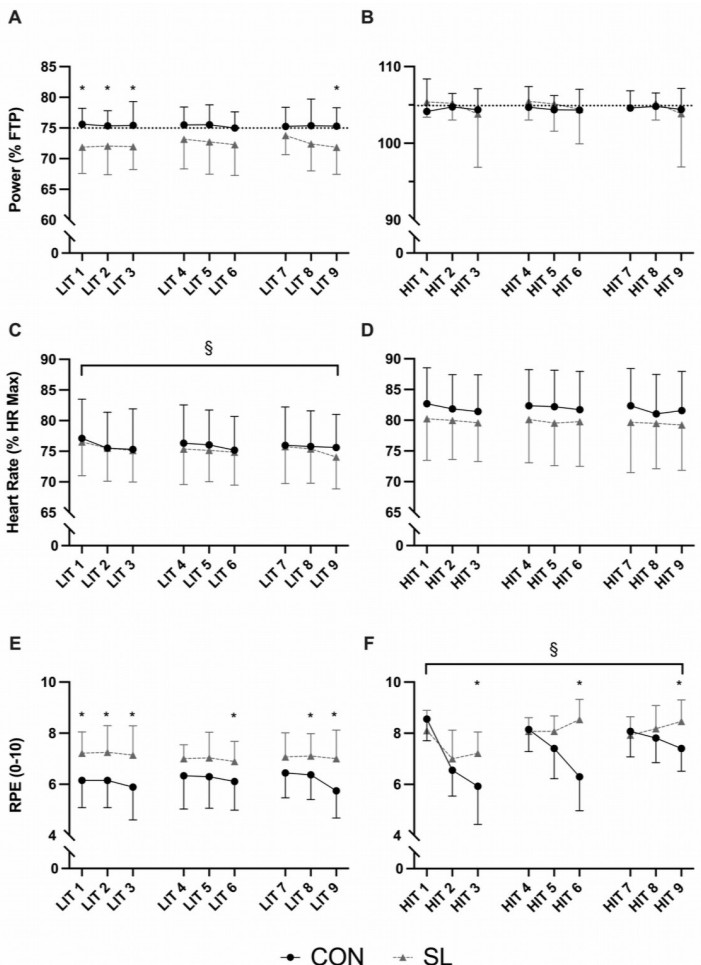

**Fig 1. Training data recorded during each low- and high-intensity training sessions (LIT and HIT, respectively).**
A, relative power output with the dotted line representing the 75% FTP target intensity for LIT sessions; B, relative power output with dotted line representing 105% FTP target intensity for HIT sessions; C, Heart rate for LIT sessions; D, Heart rate for HIT sessions; E, RPE for LIT sessions; F, RPE for HIT sessions. Grey triangles and black circles represent mean responses for sleep low (SL) and control (CON), respectively. All data are presented as mean ± SD. * denotes significant between-group difference, § denotes a significant time effect. $P < 0.05$ for all significant differences.

Mean HR during LIT sessions decreased across the intervention ($P \leq 0.01$) in similar proportions between SL (session 1: 144 ± 10; session 9: 139 ± 9 beats.min$^{-1}$, $d = 1.12$) and CON (session 1: 145 ± 14; session 9: 142 ± 14 beats.min$^{-1}$, $d = 1.05$) with no difference between groups ($P = 0.24$, $d = 0.10$) (Fig 1C). There was no time ($P = 0.19$) or between-group ($P = 0.24$, $d = 0.38$) difference in mean HR for each HIT session (Fig 1D).

RPE was significantly higher in the SL compared to CON during LIT session 1 (SL: 7.2 ± 0.8 *vs.* CON: 5.8 ± 1.5, $P < 0.01$, $d = 1.29$), session 2 (SL: 7.4 ± 1 *vs.* CON: 6.2 ± 1.1, $P < 0.01$, $d = 1.18$), session 3 (SL: 7.3 ± 1.0 *vs.* CON: 6.2 ± 1.1, $P < 0.01$, $d = 0.89$), session 6 (SL: 6.8 ± 0.8 *vs.* CON: 6.2 ± 1.1, $P = 0.04$, $d = 0.69$), and session 9 (SL: 7.0 ± 1.2 *vs.* CON: 5.8 ± 1.1, $P < 0.01$, $d = 1.06$). There was no effect of time ($P = 0.29$, $d = 0.10$) on RPE during LIT sessions (Fig 1E).

RPE was significantly higher in the SL compared to CON during HIT session 3 (SL: 7.2 ± 0.8 *vs.* CON: 5.9 ± 1.5, $P < 0.01$, $d = 1.16$), session 6 (SL: 8.5 ± 0.8 *vs.* CON: 6.4 ± 1.3, $P < 0.01$, $d = 2.04$) and session 9 (SL: 8.6 ± 0.8 *vs.* CON: 7.5 ± 0.9, $P < 0.01$, $d = 1.31$). In both

groups, RPE gradually decreased ($P < 0.01$) over HIT sessions in week 1 (Fig 1F). RPE during HIT sessions then increased (all $P < 0.01$) in SL in week 2 (from 8.1 ± 0.5 to 8.5 ± 0.6, $d = 0.43$) and week 3 (from 7.9 ± 0.7 to 8.6 ± 0.8, $d = 0.58$) whilst, in CON, it decreased (all $P < 0.01$) in a stepwise manner from 8.2 ± 0.9 to 6.5 ± 1.0 and 5.9 ± 1.5 in week 1 ($d = 1.67$), from 8.2 ± 0.9 to 6.4 ± 1.3 ($d = 1.17$) in week 2, and from 8.1 ± 1.0 to 7.5 ± 0.9 ($d = 0.46$) in week 3.

Body Mass decreased from Pre- to Post-intervention in SL (Pre: 74.7 ± 10.7; Post: 73.6 ± 10.5 kg, $P < 0.01$, $d = 0.11$) whereas no significant change was recorded in CON (Pre: 74.1 ± 9.5; Post: 74.1 ± 9.5 kg).

### Reliability of repeated home-based power tests

Reliability data for power variables and HR are reported in Table 4. There was no significant difference between test and retest for mean power during 20-min ($P = 0.69$), 5-min ($P = 0.75$) and 1-min tests ($P = 0.76$), and estimated FTP ($P = 0.69$). HR was not significantly different between test and retest in 20-min ($P = 0.49$), 5-min ($P = 0.34$) and 1-min PPO tests ($P = 0.43$).

### 20-min peak power output and functional threshold power

Both groups increased their 20-min PPO following the intervention (+4.0 ± 3.2% and +1.2 ± 3.5% in SL and CON, respectively, $P < 0.01$), with SL increasing to a greater extent (Pre: 271 ± 56; Post: 282 ± 59 W, $P < 0.01$, $d = 0.18$) compared to CON (Pre: 272 ± 54; Post: 275 ± 54 W, $P = 0.19$ $d = 0.05$) (Fig 2A). FTP was enhanced by 5.5 ± 2.6% (Pre: 3.43 ± 0.61; Post: 3.61 ± 0.63 W.kg$^{-1}$, $P < 0.01$, $d = 0.34$) in SL, whereas it was only enhanced by 1.3 ± 3.6% (from 3.49 ± 0.82 to 3.46 ± 0.8 W.kg$^{-1}$, $P = 0.10$, $d = 0.05$) in CON (Fig 2B). The HR response

**Table 4. Mean power output and heart rate data across first and second re-tests, and reliability statistics between re-tests.**

| Measures | Test 1[a] | Test 2[a] | Mean[a] | Mean Diff (W)[a] | Mean Diff (%)[a] | $d$ | CV[a] | TEM (W)[b] | TEM (CV, %)[b] | ICC[b] | $r^2$ | Bias (W)[c] |
|---|---|---|---|---|---|---|---|---|---|---|---|---|
| **20 min MPO** | | | | | | | | | | | | |
| Power (W) | 266 ± 55 | 271 ± 55 | 269 ± 55 | 4.18 ± 5.92 | 1.66 ± 2.42 | 0.08 | 1.52 ± 1.45 | 4.2 (3.6–5) | 1.7 (1.5–2.1) | 0.994 (0.991–0.996) | 0.995 | 4.182 (-7.414–15.78) |
| Heart Rate (Beats·min$^{-1}$) | 170 ± 10 | 171 ± 11 | 170 ± 10 | -0.11 ± 2.27 | -0.08 ± 1.43 | 0.01 | 0.85 ± 0.54 | 1.7 (1.5–2.1) | 1 (0.9–1.2) | 0.977 (0.963–0.986) | 0.974 | - 1.458 (-5.272–2.355) |
| **5 min MPO** | | | | | | | | | | | | |
| Power (W) | 310 ± 64 | 314 ± 65 | 312 ± 64 | 3.91 ± 7.4 | 1.23 ± 2.41 | 0.06 | 1.4 ± 1.31 | 5.2 (4.5–6.2) | 1.7 (1.5–2) | 0.994 (0.991–0.996) | 0.994 | 3.909 (-10.60–18.42) |
| Heart Rate (Beats·min$^{-1}$) | 171 ± 10 | 169 ± 11 | 170 ± 10 | 1.76 ± 3.18 | 1.21 ± 2.02 | 0.17 | 1.04 ± 1.38 | 2.4 (2–2.9) | 1.5 (1.3–2.3) | 0.944 (0.91–0.965) | 0.944 | -2.064 (-8.633–4.506) |
| **1 min MPO** | | | | | | | | | | | | |
| Power (W) | 441 ± 123 | 448 ± 127 | 445 ± 125 | 7.11 ± 22.31 | 1.42 ± 4.61 | 0.06 | 2.36 ± 2.53 | 15.8 (13.7–18.8) | 3.4 (2.9–4) | 0.989 (0.982–0.993) | 0.988 | 7.109 (-36.63–50.85) |
| Heart Rate (Beats·min$^{-1}$) | 164 ± 13 | 164 ± 13 | 164 ± 12 | -0.24 ± 4.02 | -0.19 ± 2.61 | 0.02 | 1.48 ± 1.08 | 3.1 (2.6–3.7) | 1.9 (1.6–2.3) | 0.944 (0.91–0.965) | 0.943 | -2.021 (-9.585–5.542) |
| **FTP** | | | | | | | | | | | | |
| Power (W) | 253 ± 52 | 257 ± 52 | 255 ± 52 | 3.97 ± 5.62 | 1.66 ± 2.42 | 0.08 | 1.52 ± 1.45 | 4 (3.4–4.7) | 1.7 (1.5–2.1) | 0.994 (0.991–0.996) | 0.995 | -3.973 (-14.99–7.044) |

[a] Data reported as mean ± standard deviation.

[b] Values in parentheses represent 95% Confidence intervals.

[c] Values in parenthesis represent 95% limits of agreement.

MPO, Mean power output; FTP, Functional Threshold Power; d, effect size; TEM, Typical error of the mean; CV, coefficient of variation; ICC, intraclass correlations; r$^2$, Pearson's correlation co-efficient.

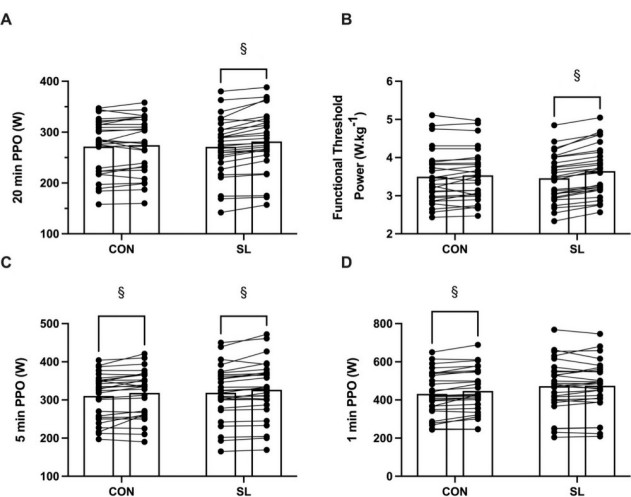

**Fig 2. Mean power output recorded during the performance tests performed before (Pre) and after (Post) the intervention.** A, Mean power output (W) during the 20-min PPO test; B, Mean functional threshold power (W·kg⁻¹); C, Mean power output (W) during the 5-min PPO test; D, Mean power output (W) during the 1-min PPO test. Bars represent means for Sleep low (SL) and control (CON) groups with individual changes represented by connected dots. § denotes a significant difference between Pre and Post. $P < 0.05$ for all significant differences.

during the 20-min PPO test was not altered following the intervention ($P = 0.19$, $d = 0.21$) both in SL (Pre: 171 ± 9; Post: 171 ± 9 beats.min⁻¹) and CON (Pre:171 ± 12; Post: 169 ± 14 beats.min⁻¹).

## 5-min peak power output

Both groups increased their 5-min PPO following the intervention (+2.3 ± 2.7% and +2.6 ± 4.9% in SL and CON, respectively, $P < 0.01$) with no significant difference between SL (Pre: 319 ± 69; Post: 327 ± 74 W, $P < 0.01$, $d = 0.10$) and CON (Pre: 311 ± 60; Post: 319 ± 62 W, $P < 0.01$, $d = 0.13$) (Fig 2C). Mean HR during the 5-min PPO test was not altered following the intervention ($P = 0.61$, $d = 0.07$) both in SL (Pre: 170 ±11; Post: 170 ± 10 beats.min⁻¹) and CON (Pre: 171 ± 11; Post: 171 ± 11 beats.min⁻¹).

## 1-min peak power output

The 1-min PPO increased by 3.9 ± 4.9% in CON following the intervention (Pre: 431.7 ± 115.0; Post: 447 ± 116 W, $P < 0.01$, $d = 0.13$), whereas it plateaued in SL (Pre: 473 ± 136; Post: 475 ± 133 W, $P = 0.31$, $d = 0.01$) (Fig 2D). Mean HR during the 1-min PPO test was not altered following the intervention ($P = 0.38$, $d = 0.12$) in SL (Pre: 166 ± 12; Post: 166 ± 12 beats.min⁻¹) or CON (Pre: 163 ± 14; Post: 164 ± 14 beats.min⁻¹).

## Nutritional intake

Mean energy intake increased significantly during training compared to baseline in SL (+300 ± 402 kcal.d⁻¹, $P < 0.01$, $d = 0.93$), whereas it was not significant in CON (+151 ± 498 kcal.d⁻¹, $P = 0.41$, $d = 0.59$). CHO intake increased during training compared to baseline in both groups ($P < 0.01$, SL: $d = 0.37$, CON: $d = 1.91$) with a concomitant reduction in fat intake ($P < 0.01$, SL: $d = 0.69$, CON: $d = 0.42$). Protein intake was significantly increased in SL ($P = 0.04$, $d = 0.67$) but did not differ between training and baseline in CON ($P = 0.38$, $d = 0.35$).

**Table 5. Mean daily energy and macronutrient intake for sleep low (SL) and control (CON) groups before the training programme (baseline) and during the training-nutrition intervention.**

| | | Energy (kcal·d⁻¹) | CHO (g·kg⁻¹·d⁻¹) | Fat (g·kg⁻¹·d⁻¹) | Protein (g·kg⁻¹·d⁻¹) |
|---|---|---|---|---|---|
| CON | Baseline | 2621 ± 485 | 3.5 ± 0.7 | 1.7 ± 0.3 | 1.7 ± 0.3 |
| | Training | 2772 ± 496 | 5.1 ± 0.9§ | 1.1 ± 0.1§ | 1.8 ± 0.3 |
| SL | Baseline | 2616 ± 495 | 3.2 ± 0.6 | 1.7 ± 0.3 | 1.7 ± 0.2 |
| | Training | 2905 ± 426§ | 5.3 ± 0.7§ | 1.1 ± 0.3§ | 1.9 ± 0.2§ |

§ denotes a significant difference from baseline, with $P < 0.05$.

There was no significant between-group difference for energy and/or macronutrient intake (Table 5).

## Discussion

The aim of the present study was to verify the effectiveness of the "Sleep low-Train low" strategy applied in "real-world" conditions outside of laboratory setting. To do this, we conducted a 3-week home-based exercise and nutrition intervention with endurance-trained athletes who performed all sessions remotely. Utilising a commercially available online coaching platform (Trainingpeaks™ LLC), we were able to characterise the day-to-day training responses of participants adhering either to a "sleep low-train low" model (SL) or control (CON), according to the design proposed by Marquet et al. [23]. Despite consistently impaired exercise intensity and higher RPE during LIT sessions, SL improved their 20-min PPO, 5-min PPO, and FTP (normalised to BM), whereas CON only improved 5-min and 1-min PPO. Overall, our results show the reliability of home-based exercise testing and provide evidence to show periodised CHO intake improves aerobic exercise performance.

To address our aim, we implemented a 3-week home-based "sleep low-train low" model where participants monitored their own training responses, performances, and dietary intake. This was possible with the participants using their own cycling power meter connected to an online training platform. Due to the expansion of power meters through reduced cost and improvements in their reproducibility [50], the implementation of power-based training prescription has become increasingly popular in cyclists over the last several years. Using this approach, coaches can consult, analyse, and monitor a range of physiological (HR, power, pace/speed, energy expenditure) and perceptual (RPE, overall feeling and wellness) training metrics for multiple athletes simultaneously [60–63]. Additionally, this approach allowed us to recruit a large number of participants (55 in our study compared to 21 and 13 in Marquet et al. [23] and Riis et al. [24] who used a similar study design, albeit laboratory based, respectively), and analyse day-to-day responses to the "sleep low-train low" intervention for the first time. Moreover, since our study was conducted during the worldwide COVID-19 pandemic (from April to June 2020), participants were subject to strict National lockdowns where the prescribed training sessions constituted their main daily physical activity. Accordingly, we are confident that the experimental conditions were similar between participants, reinforcing the validity of the data presented in this article.

In relation to performance outcomes, we observed a 4.0%, 2.3% and 5.5% improvements in 20-min PPO, 5-min PPO, and FTP (W·kg⁻¹) respectively in SL, whereas CON only improved 5-min PPO and 1-min PPO by 2.6% and 3.9%. Utilising a similar exercise-nutrition intervention, Marquet et al. [23] reported improved 10-km running time (-2.9%) and increased supra-maximal cycling (150% maximal aerobic power) time to exhaustion (+12.5%) in triathletes,

showing improvements during both aerobic and anaerobic type exercise. More recently, endurance-trained individuals subject to four weeks of "sleep low" also improved their 30-min cycling time trial performance (+19% compared to +14% in the control group) but the between-group difference was not significant [24]. This may be due to the considerably smaller sample size (n = 13) than in Marquet et al. [23] (n = 21) and our present study (n = 55), and/or differences in training load between studies as training intensities were prescribed either according to maximal HR or power output. Albeit not recorded in the present study, an improved locomotion efficiency has previously been associated with improvements in endurance performance following the "sleep low-train low" model, thus allowing athletes to sustain higher exercise intensities during physical tests [23, 64]. Given the correlation between FTP and performance during mass start cycling events, a greater improvement in FTP would be considered a desirable performance outcome for cyclists and triathletes [41]. The intensity and duration of a 20-min effort means the predominant energy substrates are endogenous CHO stores, specifically, skeletal muscle glycogen [65, 66]. Whilst increased resting muscle glycogen content is frequently observed following chronic low CHO training [3, 5, 20, 67, 68], the duration of the FTP test disallows muscle glycogen depletion as a limiting factor to performance [69]. Alternatively, augmented muscle oxidative capacity is a potential mechanism for increased 20-min PPO via improved mitochondrial function and/or biogenesis [70, 71]. It is well established within the "train-low" literature that periodically completing exercise with low muscle glycogen availability augments post-exercise cell signalling responses characterised by increased AMPK [5, 72–74], acetyl-CoA carboxylase (ACC) activity [5] and p53 signalling [3]. An augmented molecular response to exercise under conditions of low muscle glycogen leads to elevated PGC-1α mRNA expression [3–5, 17, 74], and nuclear translocation [75]. Furthermore, the chronic application of "train-low" has shown an amplified adaptive response with increased markers of mitochondrial adaptation such as citrate synthase, OXPHOS subunit COX IV [20], 3-hydroxyacyl-CoA dehydrogenase (β HAD) [22, 67, 71], and succinate dehydrogenase (SDH) [27] activities suggestive of improved mitochondrial efficiency [71]. Utilising an alternative CHO periodisation strategy (twice daily), Cochran et al. [76] showed improved performance following two weeks of training with low CHO availability, despite no changes in mitochondrial protein content, speculating that improved performance may be due to changes in efficiency, which can occur independent of changes in content [77]. Considering the acute molecular responses to training with low CHO availability, and the improvements in mitochondrial efficiency following the chronic application of "train-low", it is justified to speculate improvements in muscle oxidative capacity leads to increased performance following a period of low CHO training. However, the placebo effect of the "sleep low-train low" model cannot be excluded and may also explain part of the performance improvement [78].

The lack of improvement to the 1-min PPO test in SL contrasts with evidence from previous studies [23, 79] and may be explained by several factors. Firstly, the lack of supervision during this supra-maximal testing session could have led to a sub-optimal execution (as reflected by a higher CV compared to the 5-min and 20-min tests) as it is not often performed by endurance athletes [78]. Secondly, a mechanistic rationale exists, whereby pyruvate dehydrogenase kinase 4 (PDK4) expression is increased following a period of "train low" leading to alterations in mitochondrial CHO flux. PDK4 is an exercise sensitive gene which encodes the protein PDK [80] and mRNA expression is increased following a single bout of "sleep low-train low" [2–5]. Crucially, PDK inhibits pyruvate dehydrogenase (PDH), a critical enzyme complex in the regulation of substrate metabolism during exercise. It remains to be reasoned that decreased PDH activity would impair CHO flux into the mitochondria, increasing lactate production, and limiting performance at supra-maximal exercise intensities [81]. This rationale has been used to justify impaired high-intensity sprint performance following the use of

low CHO, high fat (LCHF) nutritional interventions [82–84] and may be similarly responsible for impaired 1-min PPO in SL. Contrastingly, a previous laboratory-based study utilising the same study design has shown improvements in supramaximal cycling performance following as little as 1 week [64] and 3 weeks of sleep low-train low [23]. This highlights the importance of completing further research to gain greater insight into the performances effect of periodically training with low CHO availability, meaning coaches and nutritionists must consider when to implement low CHO training during an athlete's training programme [28].

Understanding the daily impact of training with reduced CHO availability is important for coaches, sport scientists and nutritionists alike. Hulston et al. [22] and Yeo et al. [20], have previously reported the effects of training with low CHO availability on HIT session power output across a "train low" programme. The former reported reduced power output was across all HIT sessions whilst Yeo and colleagues showed reduced power output across the first week in the CHO group, trending towards similar exercise intensity between groups after 3 weeks of intervention. In these two studies, participants completed LIT sessions with high CHO, with subsequent HIT sessions with low CHO availability, resulting in less work done under low CHO conditions and compromised training quality. As with Marquet et al. [23] and in line with the fuel for the work required paradigm [85], the present study aimed to align CHO intake with exercise demand with the purpose of augmenting training adaptation during fasted LIT sessions, extending the period of low CHO availability overnight and providing high CHO availability during HIT sessions to ensure session completion as prescribed. Whilst there was no difference between groups in mean HR, the data is clear when considering exercise intensity, and the perceptual effect of training with low CHO availability. Despite being asked to maintain 75% FTP during the LIT sessions, SL participants were consistently below target power. Typically, endurance exercise undertaken with critically low muscle glycogen concentrations ($< 200$ mmol.kg dw$^{-1}$) results in impaired exercise capacity due to lacking muscle energy substrates and impaired contractile capacity via compromised calcium regulation [86–88]. Evidence from our laboratory has shown that the provision of a CHO-free sweet placebo drink could partially restore exercise capacity during a "sleep low-train low" approach [89]. Progressing this further, providing exogenous CHO ingestion during exercise under low CHO conditions does not impair fat oxidation during exercise [90, 91] and therefore, providing a moderate dose of CHO may serve to improve exercise capacity by preventing hypoglycaemia during exercise and recovery. However, in our study, since HR data during LIT sessions were similar between SL and CON, we can hypothesise the intensity was sufficiently difficult for SL despite a reduction in power output. Moreover, RPE during LIT sessions were consistently higher in SL compared to CON, which could contribute to better fatigue resistance, amplifying the metabolic benefits of the session. By periodising CHO effectively, we successfully restored exercise intensity during HIT sessions, allowing both groups to perform the sessions as prescribed. This was evident by the lack of difference between SL and CON across all intervals and sessions for power output, HR and RPE. Taken together, our performance data confirm that the strategic periodisation of CHO around training sessions, with HIT sessions performed with high CHO availability and LIT sessions performed with low CHO availability, may confer an advantage to endurance athletes. A high CHO availability for HIT sessions allows the maintenance of a higher training intensity, but it remains to be understood whether completing HIT exercise with deliberately low muscle glycogen positively alters the metabolic milieu within the muscle, even at the expense of absolute workload.

In relation to the nutritional intervention, participants in both groups increased their CHO intake during the "sleep low-train low" intervention (5.2 and 5.1 g·kg$^{-1}$·d$^{-1}$ in SL and CON, respectively) compared to their habitual diet (3.1 and 3.4 g·kg$^{-1}$·d$^{-1}$ in SL and CON, respectively). Despite this increase in CHO intake, all participants failed to reach the prescribed 6

g·kg$^{-1}$·d$^{-1}$, a similar finding to Marquet et al. [23], where participants' CHO intake was 5.4 and 5.6 g·kg$^{-1}$·d$^{-1}$ in SL and CON, respectively. A possible reason for this is that we attempted to achieve the CHO intake through food ingestion and did not include any high CHO drink or gel supplementation as is often provided in acute CHO periodisation studies to maximise refuelling between sessions [32]. This provides useful practical evidence that without provision of high CHO supplements, athletes may struggle to achieve the high CHO intake required between sessions to replenish muscle glycogen stores. Furthermore, in SL, CHO ingestion was only permitted in the hours between LIT and HIT sessions providing a relatively small window within which to consume CHO. Despite consuming below the recommended CHO intake, SL sufficiently restored CHO to maintain the required exercise intensity during HIT session (105% FTP), as inferred by the lack of difference between SL and CON for power output and HR. This result also suggests a higher daily CHO intake ($\geq 6$ g·kg$^{-1}$·d$^{-1}$) may be unnecessary during a "sleep low-train low" bout in endurance athletes. Moreover, it cannot be excluded the underconsumption of CHO throughout the study may be due to under reporting nutritional intake, an issue well known when collecting food diaries in athletic populations [92], and/or the use of a nutrition app as a tool to record dietary intake may also lead to underestimation of macronutrient intake [93]. Despite this, there is no reason to believe the likelihood of underreporting would be greater in either group, therefore comparisons between groups remain valid. To mitigate this risk, all participants were provided with comprehensive, written dietary guidelines with macronutrient compositions for all meals and snacks, and volumes and quantities of all drinks/foods required to achieve the desired dietary intakes.

Using technology to facilitate exercise monitoring, prescription, and testing, we have shown the potential for data collection away from traditional laboratory-based testing and supervision. The tests selected were familiar to athletes and coaches and therefore directly transferable to practice, through the alignment of common testing methodologies and the lack of need for expensive laboratory equipment. Despite this, the best approach for research-centric interventions may be to combine online training and nutritional prescription, with laboratory-based testing protocols to ensure optimal reliability and ensure subject safety during extreme exercise tests. Exercise can be completed in a semi-supervised manner using online coaching platforms, allowing researchers to prioritise Pre-Post testing athletes in the laboratory, and alleviating strain on laboratory resources and space. Similarly, the requirement for participants to attend the laboratory for testing sessions only, as opposed to multiple times weekly to complete a training programme, may facilitate greater participant numbers in chronic exercise training interventions. Despite this, a potential limitation of remote data collection is ensuring that standardisation and quality control is maintained across all participants. Clear instructions must be given of the necessary protocols to ensure robust data collection. In the present study, all participants were given the same instructions but were encouraged to contact the research team if they were unsure or required any assistance with understanding the written instructions. Furthermore, to ensure only correctly collected data was used in the study, only participants with 100% compliance rates across all tests and training sessions were included in the analysis. Accordingly, in future protocols we recommend that all training sessions and tests be visually inspected (i.e. review power files) to ensure that the protocols are followed correctly.

## Practical application

The practical implications of the present study are twofold. Firstly, home-based performance testing represents a highly reliable strategy for athletes and coaches to monitor longitudinal training programme efficacy. With appropriate standardisation of diet, exercise and physical

activity prior to performance tests, coaches and athletes can expect ~2% variation between repeated trials, and any greater difference between repeated trials can be attributed to variables outside of the inherent variability of these tests. Secondly, the characterisation of exercise capacity and performance outcomes following a period of training with low-CHO availability highlight, despite reductions in exercise intensity, that performance outcomes are still improved. Coaches and athletes should be aware that training with low endo- and exogenous CHO availability impairs exercise capacity, but exercise intensity can be restored on the same day providing adequate CHO is provided between exercise bouts.

## Conclusion

Despite reductions in relative training intensity, we provide data that demonstrates three weeks of "sleep low-train low" is effective to improve functional threshold power (FTP) and 5-min PPO in trained cyclists and triathletes, with no benefit to high-intensity exercise performance (1-min PPO) compared to "normal" carbohydrate availability. A major novelty of this study is the use of remote exercise and nutrition prescription and data collection during a home-based intervention (completed throughout COVID-19 pandemic). In this way, our data support the feasibility of implementing a "sleep low-train low" intervention outside of typical laboratory-controlled conditions.

## Acknowledgments

We would like to thank all participants for their commitment to the study through uncertain times.

## Author Contributions

**Conceptualization:** Samuel Bennett, Eve Tiollier, Franck Brocherie, Daniel J. Owens, James P. Morton, Julien Louis.

**Data curation:** Samuel Bennett, Julien Louis.

**Formal analysis:** Samuel Bennett, Julien Louis.

**Investigation:** Samuel Bennett.

**Methodology:** Samuel Bennett, Eve Tiollier, Franck Brocherie, Daniel J. Owens, James P. Morton, Julien Louis.

**Supervision:** Samuel Bennett, Eve Tiollier, Franck Brocherie, Daniel J. Owens, James P. Morton, Julien Louis.

**Writing – original draft:** Samuel Bennett, Julien Louis.

**Writing – review & editing:** Samuel Bennett, Eve Tiollier, Franck Brocherie, Daniel J. Owens, James P. Morton, Julien Louis.

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
