## [Decision Letter · Decision Letter 0]

15 Sep 2021

PONE-D-21-20297Three weeks of a home-based “sleep low-train low” intervention improves functional threshold power in trained cyclists: a feasibility study undertaken during the Covid-19 pandemic.PLOS ONE

Dear Dr. Julien Louis,

Thank you for submitting your manuscript to PLOS ONE. After careful consideration, we feel that it has merit but does not fully meet PLOS ONE’s publication criteria as it currently stands. Therefore, we invite you to submit a revised version of the manuscript that addresses the points raised during the review process.

We look forward to receiving your revised manuscript.

Kind regards,

Cosme F. Buzzachera, Ph.D.

Academic Editor

PLOS ONE

Reviewers' comments:

Reviewer's Responses to Questions

**Comments to the Author**

1. Is the manuscript technically sound, and do the data support the conclusions?

Reviewer #1: Partly

Reviewer #2: Yes

2. Has the statistical analysis been performed appropriately and rigorously? 

Reviewer #1: Yes

Reviewer #2: Yes

3. Have the authors made all data underlying the findings in their manuscript fully available?

Reviewer #1: No

Reviewer #2: Yes

4. Is the manuscript presented in an intelligible fashion and written in standard English?

Reviewer #1: Yes

Reviewer #2: Yes

5. Review Comments to the Author

Reviewer #1: PONE-D-21-20297 Three weeks of a home-based “sleep low-train low” intervention improves functional threshold power in trained cyclists: a feasibility study undertaken during the Covid-19 pandemic.

The authors are commended for a well-written manuscript. The arguments for the manuscript under review are timely and original. There are, however, significant concerns and methodological issues with the manuscript in its current form, and in my view, the document is unsuitable for publication. The fundamental flaw is the use of diverse, non-scientifically validated power meters. This situation is valid in practical terms but not adequate from a rigid, scientific perspective. All my comments are included below. I hope you will find them to be constructive and helpful.

The primary concerns with the manuscript are presented below:

An unfortunate aspect of the study is the use of multiple, non-scientifically validated (not all!) power meters. This fundamental flaw is vital since power output (PO) data are crucial for the current findings. Although the authors have reported the manufacturer's claimed power meter accuracy, it is not valid from a rigid, scientific perspective. Both validity and reliability of an ergometer/power meter are necessary within a scientific context. Why? Variability within an ergometer (or similar) could arise from both systematic and random error sources. Both error sources, for example, limit our ability to compare repeated tests over time or track changes in physiological outcomes (for details, please see Hopker et al., 2010). Thus, were the improvements in FTP derived from the nutritional intervention per se or related to systematic/random error sources of such devices? Please comment. Lastly, if the calibration was done (lines 204-206), was it done using a dynamic calibration rig - or similar? Please explain.

Hopker J, et al. Validity and reliability of the Wattbike cycle ergometer. Int J Sports Med. 2010.

Some sample characteristics were poorly reported. For example, were the participants in good health or users of any supplement and/or medication that could affect their metabolism - and indirectly, performance? Also, how were the participants recruited? This kind of information is crucial for the readers. Please insert. Lastly, the inclusion of sample size calculation is also essential for the reader. Is the actual sample (n = 55) enough to explore the hypotheses of the current investigation (I believe yes!)? If not, the reduced sample size should be acknowledged as a limitation. Please insert sample size calculation.

Session RPE was assessed by the 0–10 Borg RPE scale. This relatively simple scale, however, requires a multitude of procedures, including the standard definition of perceived exertion and memory- and/or exercise-anchoring procedures (for details, see Noble & Robertson, 1996). Additionally, to define the assessment time following a training session or test (15-min, 30-min, 1-h, 2-h,...) is also essential. Therefore, the lack of these RPE-related procedures should be acknowledged as a limitation.

Noble BJ, Robertson RJ. Perceived Exertion. Champaign: Human Kinetics. 1996.

The authors defined HR as an outcome of interest. However, some pieces of information are still unclear. For example, how was HR monitored? Please specify (Polar, Garmin, Suntuu,...). Importantly, what means "mean HR"? HR is highly variable during a training session and depends on multiple factors other than the effort per se (emotional stress, body temperature, hydration status, use of medications, and others). Hence, the "mean" HR may not reflect the physiological strain of a session training. In addition, how was HRmax estimated or measured? Please inform.

Although the Ethics committee has approved the study, the authors should consider the risks of the proposed study protocol. Today, exercise physiology labs worldwide are spending much money to increase the safety of both evaluators and volunteers. In the current study, however, each volunteer was asked to perform three heavy tests without any in loco supervision. I understand that this lack of supervision is related to the pandemic and that the participants were "trained" adults. However, adverse events may occur occasionally - we have a recent example in the Euro soccer championship -, and supervision is fundamental. In other words, this lack of in-person supervision in such heavy tests - or proposed training - may lead the participants to "unnecessary" risks. Therefore, sentences such as "Using technology to facilitate exercise monitoring, prescription, and testing, we have shown the potential for data collection away from traditional laboratory-based testing and supervision", should be avoided. Please consider.

The minor concerns with the manuscript are presented below:

The purpose of the study was "to assess the feasibility of a 3-week home-based “sleep low-train low” programme and its effects on cycling performance in trained athletes". On most occasions (including the title!), however, the authors reinforce the importance of the COVID-19 period. Why? There is, in my view, not necessary. The current findings could be extrapolated to any context where "outdoor" cycling training is not possible - including, but not exclusive to, the COVID-19 pandemic. The authors are encouraged, therefore, to avoid "highlighting" this specific scenario.

The study hypothesis is a bit unclear. The authors hypothesized that "performance will be improved to a greater extent in the SL group compared to control conditions". However, the authors also hypothesized that "in the absence of live supervision, adherence will remain high, with only those with 100% completion considered for further analysis". The authors did not measure adherence since they merely decided to include participants who completed all training sessions. Please exclude this final sentence.

Some pieces of information exist but are still unclear. For example, were the participants asked to avoid caffeinated or alcoholic products and vigorous exercise before (24 h) the assessments? If yes, please inform.

The Discussion section is long and exceptionally speculative, with arguments that are out of the scope of the study. An example is paragraph 3. The authors stated that "considering the acute molecular responses to training with low CHO availability, and the improvements in mitochondrial efficiency following the chronic application of “train low”, it is justified to speculate improvements in muscle oxidative capacity leads to increased performance following a period of low CHO training"; however, it is hard to speculate that this short-time of home-based training intervention improved oxidation muscle oxidative capacity w/o any direct muscle measurement. Therefore, the authors are recommended to only focus on the current findings.

Reviewer #2: Overview

The subject is interesting and has a lot of relevance, especially at this pandemic moment.

It has great applicability and shows the benefit of using remote monitoring systems.

I did some commentaries in the text (PDF file).

Introduction

At the final of this section it seems to be a scientific gap, but it´s not clear. Please revise and insert the scientific gap of the study.

Discussion

As reviewer I suggest rewrite the 1st paragraph of this section. Please observe my commentaries on PDF file.

Before conclusion I suggest insert clearly the practical application because the most important contribution of this study, in my point of view, is the applicability and benefit of using remote monitoring systems especially in this moment of pandemic, as a tool for monitoring distance training, generating reliable results and making athletes and practitioners not feel alone in relation to their training.

6. PLOS authors have the option to publish the peer review history of their article (what does this mean?). If published, this will include your full peer review and any attached files.

Reviewer #1: No

Reviewer #2: No

---

## [Author Response · Author response to Decision Letter 0]

25 Sep 2021

Please find our point-by-point response to editor and reviewers in the attached file named "response to reviewers".

---

## [Decision Letter · Decision Letter 1]

22 Nov 2021

Three weeks of a home-based “sleep low-train low” intervention improves functional threshold power in trained cyclists: a feasibility study

PONE-D-21-20297R1

Dear Dr. JULIEN LOUIS,

We’re pleased to inform you that your manuscript has been judged scientifically suitable for publication and will be formally accepted for publication once it meets all outstanding technical requirements.

Kind regards,

Cosme F. Buzzachera, Ph.D.

Academic Editor

PLOS ONE

Reviewers' comments:

Reviewer's Responses to Questions

**Comments to the Author**

1. If the authors have adequately addressed your comments raised in a previous round of review and you feel that this manuscript is now acceptable for publication, you may indicate that here to bypass the “Comments to the Author” section, enter your conflict of interest statement in the “Confidential to Editor” section, and submit your "Accept" recommendation.

Reviewer #2: All comments have been addressed

2. Is the manuscript technically sound, and do the data support the conclusions?

Reviewer #2: Yes

3. Has the statistical analysis been performed appropriately and rigorously? 

Reviewer #2: Yes

4. Have the authors made all data underlying the findings in their manuscript fully available?

Reviewer #2: Yes

5. Is the manuscript presented in an intelligible fashion and written in standard English?

Reviewer #2: Yes

6. Review Comments to the Author

Reviewer #2: The subject is interesting and has a lot of relevance, especially at this pandemic moment. It has great applicability and shows the benefit of using remote monitoring systems. The article is in conformity for publication.

7. PLOS authors have the option to publish the peer review history of their article (what does this mean?). If published, this will include your full peer review and any attached files.

Reviewer #2: No

---

## [Editor Report · Acceptance letter]

24 Nov 2021

PONE-D-21-20297R1 

Three weeks of a home-based “sleep low-train low” intervention improves functional threshold power in trained cyclists: a feasibility study 

Dear Dr. Louis:

I'm pleased to inform you that your manuscript has been deemed suitable for publication in PLOS ONE. Congratulations! Your manuscript is now with our production department. 

Kind regards, 

on behalf of

Dr. Cosme F. Buzzachera 

Academic Editor

PLOS ONE